# SELF-SUPERVISED LEARNING OF COMPRESSED VIDEO REPRESENTATIONS

**Youngjae Yu⋆, Sangho Lee⋆, Gunhee Kim**
Seoul National University
{yj.yu,sangho.lee}@vision.snu.ac.kr, gunhee@snu.ac.kr

**Yale Song**
Microsoft Research
yalesong@microsoft.com

## ABSTRACT

Self-supervised learning of video representations has received great attention. Existing methods typically require frames to be *decoded* before being processed, which increases compute and storage requirements and ultimately hinders large-scale training. In this work, we propose an efficient self-supervised approach to learn video representations by eliminating the expensive decoding step. We use a three-stream video architecture that encodes I-frames and P-frames of a compressed video. Unlike existing approaches that encode I-frames and P-frames individually, we propose to *jointly* encode them by establishing bidirectional dynamic connections across streams. To enable self-supervised learning, we propose two pretext tasks that leverage the multimodal nature (RGB, motion vector, residuals) and the internal GOP structure of compressed videos. The first task asks our network to predict zeroth-order motion statistics in a spatio-temporal pyramid; the second task asks correspondence types between I-frames and P-frames after applying temporal transformations. We show that our approach achieves competitive performance on compressed video recognition both in supervised and self-supervised regimes.

## 1 INTRODUCTION

There has been significant progress on self-supervised learning of video representations. It learns from unlabeled videos by exploiting their underlying structures and statistics as free supervision signals, which allows us to leverage large amounts of videos available online. Unfortunately, training video models is notoriously difficult to scale. Typically, practitioners have to make trade-offs between *compute* (decode frames and store them as JPEG images for faster data loading, but at the cost of large storage) and *storage* (decode frames on-the-fly at the cost of high computational requirements). Therefore, large-batch training of video models is difficult without high-end compute clusters. Although these issues are generally applicable to any video-based scenarios, they are particularly problematic for self-supervised learning because large-scale training is one key ingredient (Brock et al., 2019; Clark et al., 2019; Devlin et al., 2019) but that is exactly where these issues are aggravated.

Recently, several approaches demonstrated benefits of compressed video recognition (Zhang et al., 2016; Wu et al., 2018; Shou et al., 2019; Wang et al., 2019b). Without ever needing to decode frames, these approaches can alleviate compute and storage requirements, e.g., resulting in 3 to 10 times faster solutions than traditional video CNNs at a minimal loss on accuracy (Wu et al., 2018; Wang et al., 2019b). Also, motion vectors embedded in compressed videos provide a free alternative to optical flow which is compute-intensive; leveraging this has been shown to be two orders of magnitude faster than optical flow-based approaches (Shou et al., 2019). However, all the previous work on compressed video has focused on supervised learning and there has been no study that shows the potential of compressed videos in self-supervised learning; this is the focus of our work.

---

⋆Equal Contribution

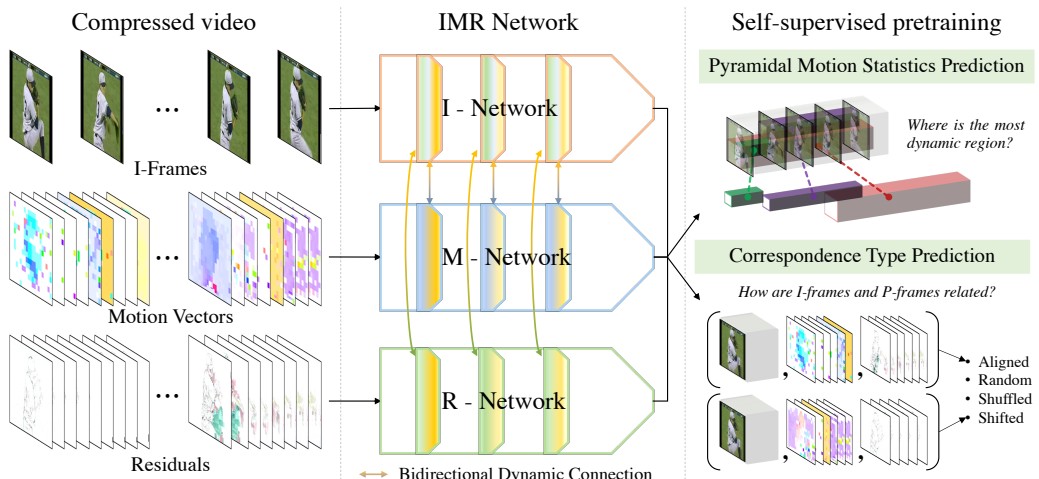

Figure 1: **IMR network** consists of three sub-networks encoding different information streams provided in compressed videos. We incorporate bidirectional dynamic connections to facilitate information sharing across streams. We train the model using two novel pretext tasks designed by exploiting the underlying structure of compressed videos.

In this work, we propose a self-supervised approach to learning video representations directly in the compressed video format. We exploit two inherent characteristics of compressed videos: First, video compression packs a sequence of images into several Group of Pictures (GOP). Intuitively, the GOP structure provides *atomic representation of motion*; each GOP contains images with just enough scene changes so a video codec can compress them with minimal information loss. Because of this atomic property, we enjoy less spurious, more consistent motion information at the GOP-level than at the frame-level. Second, compressed videos naturally provide multimodal representation (*i.e.* RGB frames, motion vectors, and residuals) that we can leverage for multimodal correspondence learning. Based on these, we propose two novel pretext task (see Fig. 1): The first task asks our model to predict zeroth-order motion statistics (*e.g.* where is the most dynamic region) in a pyramidal spatio-temporal grid structure. The second involves predicting correspondence types between I-frames and P-frames after temporal transformation. Solving our tasks require *implicitly* locating the most salient moving objects and matching their appearance-motion correspondences between I-frames and P-frames; this encourages our model to learn discriminative representation of compressed videos.

A compressed video contains three streams of multimodal information – *i.e.* RGB images, motion vectors, and residuals – with a dependency structure between an I-frame stream and the two P-frame streams punctuated by GOP boundaries. We design our architecture to encode this dependency structure; it contains one CNN encoding I-frames and two other CNNs encoding motion vectors and residuals in P-frames, respectively. Unlike existing approaches that encode I-frames and P-frames individually, we propose to *jointly* encode them to fully exploit the underlying structure of compressed videos. To this end, we use a three-stream CNN architecture and establish *bidirectional dynamic connections* going from each of the two P-frame streams into the I-frame stream, and vice versa, and put these connections layer-wise to learn the correlations between them at multiple spatial/temporal scales (see Fig. 1). These connections allow our model to fully leverage the internal GOP structure of compressed videos and effectively capture atomic representation of motion.

In summary, our main contributions are two-fold: (1) We propose a three-stream architecture for compressed videos with bidirectional dynamic connections to fully exploit the internal structure of compressed videos. (2) We propose novel pretext tasks to learn from compressed videos in a self-supervised manner. We demonstrate our approach by pretraining the model on Kinetics-400 (Kay et al., 2017) and finetuning it on UCF-101 (Soomro et al., 2012), HMDB-51 (Kuehne et al., 2011). Our model achieves new state-of-the-art performance in compressed video classification tasks in both supervised and self-supervised regimes, while maintaining a similar computational efficiency as existing compressed video recognition approaches (Wu et al., 2018; Shou et al., 2019).

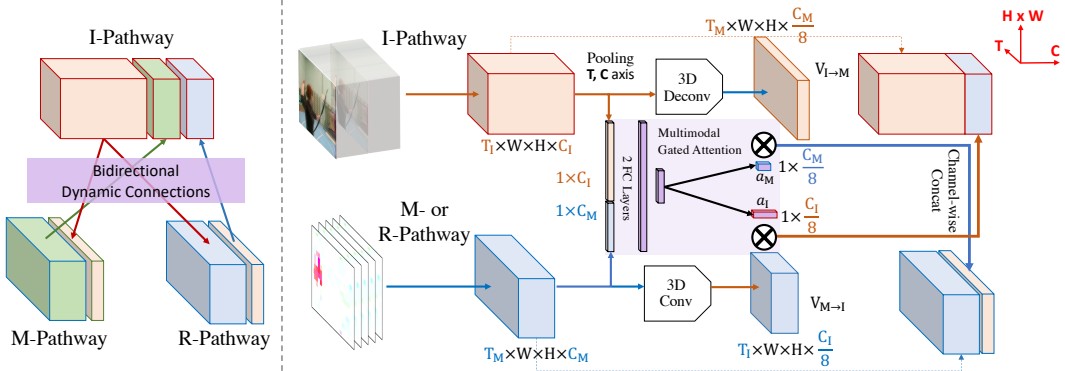

Figure 2: **Bidirectional dynamic connections** are established between I & M pathways and between I & R pathways to facilitate information sharing between I-frames and P-frames. We incorporate *multimodal-gated attention* to dynamically modulate the connections based on input. Feature tensors (orange and blue cubes) are placed in the T-HxW-C plane.

## 2 APPROACH

We use videos compressed according to the MPEG-4 Part 2 specifications (Le Gall, 1991) as our input, following the previous work (Wu et al., 2018; Shou et al., 2019; Wang et al., 2019b). This compression format encodes an RGB image sequence as a series of GOPs (Group of Pictures) where each GOP starts with one I-frame followed by a variable number of P-frames. An I-frame stores RGB values of a complete image and can be decoded on its own. A P-frame holds only the changes from the previous reference frame using motion vectors and residuals. The motion vectors store 2D displacements of the most similar patches between the reference and the target frames, and the residuals store pixel-wise differences to correct motion compensation errors. We use all the three modalities contained in compressed videos as our input.

Formally, our input is $T$ GOPs, $G_0, \cdots, G_{T-1}$, where each $G_t$ contains one I-frame $I_t \in \mathbb{R}^{H \times W \times 3}$ followed by $K - 1$ pairs of motion vectors $M_{t,k} \in \mathbb{R}^{H \times W \times 2}$ and residuals $R_{t,k} \in \mathbb{R}^{H \times W \times 3}$, $k \in [1, K)$. For efficiency and simplicity, we assume an identical GOP size $K$ for all $t \in [0, T)$.

### 2.1 IMR NETWORK FOR COMPRESSED VIDEOS

Our model consists of three CNNs, each with 3D convolutional kernels modeling spatio-temporal dynamics within each input stream $\{I_t\}, \{M_{t,k}\}, \{R_{t,k}\}, t \in [0, T), k \in [0, K)$; we denote these sub-networks by I-network $f_I$, M-network $f_M$, and R-network $f_R$, respectively, and call our model *IMR Network (IMRNet)*. We account for the difference in the amount of information between I-frames and P-frames by adjusting the capacity of networks accordingly. Specifically, following (Wu et al., 2018), we make the capacity of $f_I$ larger than $f_M$ and $f_R$ by setting the number of channels in each layer of $f_I$ to be $\gamma$ times higher than those of $f_M$ and $f_R$ (we set $\gamma = 64$).

Existing models for compressed videos typically perform late fusion (Wu et al., 2018; Shou et al., 2019), i.e., they combine embeddings of I-frames and P-frames only *after encoding* each stream. However, we find that it is critical to allow our sub-networks to share information *as they encode* their respective input streams. To this end, we establish layer-wise lateral connections between $f_I$ & $f_M$ and between $f_I$ & $f_R$.

**Bidirectional dynamic connections.** Lateral connections have been used to combine information from different streams, e.g., RGB images and optical flow images (Feichtenhofer et al., 2016), and RGB images sampled at different frame rates (Feichtenhofer et al., 2019). In this work, we use it to combine information from I-frames and P-frames. Our approach is different from previous work in two key aspects: (1) We establish *bidirectional* connections between streams, instead of unidirectional connections as was typically done in the past (Feichtenhofer et al., 2016; 2019), so that information sharing is symmetrical between streams. (2) We incorporate *multimodal gated attention* to dynamically adjust the connections based on multimodal (I-frame and P-frames) information. We call our approach *bidirectional dynamic connections* to highlight these two aspects and differentiate

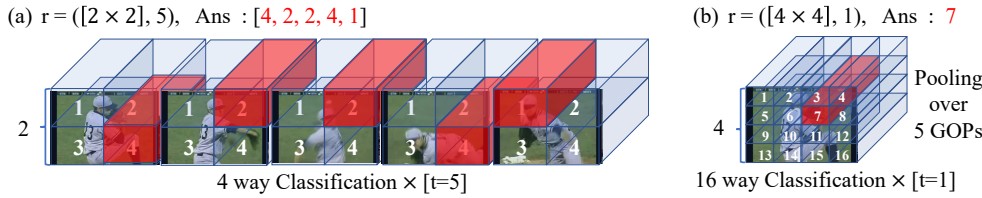

(a) r = ([2 × 2], 5),  Ans : [4, 2, 2, 4, 1]    (b) r = ([4 × 4], 1),  Ans : 7

Figure 3: **Pyramidal motion statistics prediction** asks our network to find a region with the highest energy of motion. Here we visualize two levels in a spatio-temporal pyramid for illustration.

ours from previous work, e.g., SlowFast networks (Feichtenhofer et al., 2019) establish *unidirectional* lateral connections and the connections are *static* regardless of the content from the other stream.

We combine embeddings from different sub-networks via channel-wise concatenation, which requires embeddings to match their spatio-temporal dimensions. However, $f_I$ processes $\kappa$ times less frames than $f_M$ and $f_R$, producing embeddings that are $\kappa$ times smaller in the temporal dimension. Therefore, we transform the embeddings with time-strided 3D (de-)convolution with $(\kappa \times 1 \times 1)$ kernels, $C/8$ channels, and $(\kappa, 1, 1)$ temporal stride: We use convolution for $f_I \rightarrow f_M/f_R$ to decrease the time dimension and deconvolution for $f_M/f_R \rightarrow f_I$ to increase it. Note that simply using the (de-)conv layers will perform *static* transformation regardless of what is provided from the other sub-network, similar to (Feichtenhofer et al., 2019). However, we find it critical to make the transformations aware of information from both sub-networks so that the networks can dynamically adjust the connections and selectively share only the most relevant information from each sub-network.

To achieve this, we dynamically modulate (de-)conv layer outputs using multimodal-gated attention weights. Let $\mathbf{x}_I \in \mathbb{R}^{T_I \times W \times H \times C_I}$ and $\mathbf{x}_M \in \mathbb{R}^{T_M \times W \times H \times C_M}$ be the embeddings from $f_I$ and $f_M$, respectively. We max-pool $\mathbf{x}_I$ and $\mathbf{x}_M$ and concatenate them to obtain multimodal embedding $\mathbf{z} \in \mathbb{R}^{C_z}$ with $C_Z = C_I + C_M$. We define multimodal gate functions that take as input $\mathbf{z}$ and generate attention weights $\mathbf{a}_I \in \mathbb{R}^{C_I/8}$ and $\mathbf{a}_M \in \mathbb{R}^{C_M/8}$ as

$$\mathbf{a}_I = \sigma\left(W_3\mathbf{h} + b_3\right), \ \mathbf{a}_M = \sigma\left(W_4\mathbf{h} + b_4\right), \ \mathbf{h} = \zeta\left(W_2\zeta\left(W_1\mathbf{z} + b1\right) + b2\right) \tag{1}$$

where $\sigma$ is a sigmoid function, $\zeta$ is a Leaky ReLU function, and $W_1, W_2 \in \mathbb{R}^{C_Z \times C_Z}, b_1, b_2 \in \mathbb{R}^{C_Z}, W_3 \in \mathbb{R}^{C_I/8 \times C_Z}, b_3 \in \mathbb{R}^{c_I/8}, W_4 \in \mathbb{R}^{C_M/8 \times C_Z}, b_4 \in \mathbb{R}^{C_M/8}$ are weight parameters. Next, we use these attention weights to modulate the (de-)conv output embeddings,

$$\mathbf{v}_{I \rightarrow M} = \mathbf{a}_M \otimes \texttt{3d\_conv}(\mathbf{x}_I), \ \mathbf{v}_{M \rightarrow I} = \mathbf{a}_I \otimes \texttt{3d\_deconv}(\mathbf{x}_M) \tag{2}$$

where $\otimes$ is channel-wise multiplication. We repeat the same process for $f_I$ & $f_R$ to obtain $\mathbf{v}_{I \rightarrow R}$ and $\mathbf{v}_{R \rightarrow I}$, and combine them with the feature embeddings via channel-wise concatenation,

$$\hat{\mathbf{x}}_I = [\mathbf{x}_I; \mathbf{v}_{M \rightarrow I}; \mathbf{v}_{R \rightarrow I}], \ \hat{\mathbf{x}}_M = [\mathbf{x}_M; \mathbf{v}_{I \rightarrow M}], \ \hat{\mathbf{x}}_R = [\mathbf{x}_R; \mathbf{v}_{I \rightarrow R}] \tag{3}$$

Each of these is fed into the next layer in the corresponding sub-network. We establish these lateral connections across multiple layers of our network. To obtain the final embedding, we apply average pooling on the output from the final layer of each sub-network and concatenate them channel-wise.

Note that the design of IMRNet is orthogonal to the design of video CNNs; while we adapt 3D-ResNet (He et al., 2016) as the backbone in our experiments, we can use any of existing CNN architectures as the backbone, e.g., C3D (Tran et al., 2015), I3D (Carreira & Zisserman, 2017), R(2+1)D (Tran et al., 2018). What is essential, however, is that (i) there are three sub-networks, each modeling one of the three input streams, and (ii) information from different networks are combined via bidirectional dynamic connections as they are encoded.

## 2.2 Self-Supervised Learning Objectives

Compressed videos have unique properties, i.e., the multimodal nature of information (RGB, motion vector, residuals) and the internal GOP structure that provides atomic representation of motion. We turn these properties into free self-supervisory signals and design two novel pretext tasks.

**Pyramidal Motion Statistics Prediction (PMSP).** One important desideratum of video CNNs is learning visual representation that captures salient objects and motion. We hypothesize that there is an implicit *videographer bias* captured in videos in-the-wild that naturally reflect visual saliency: Videos

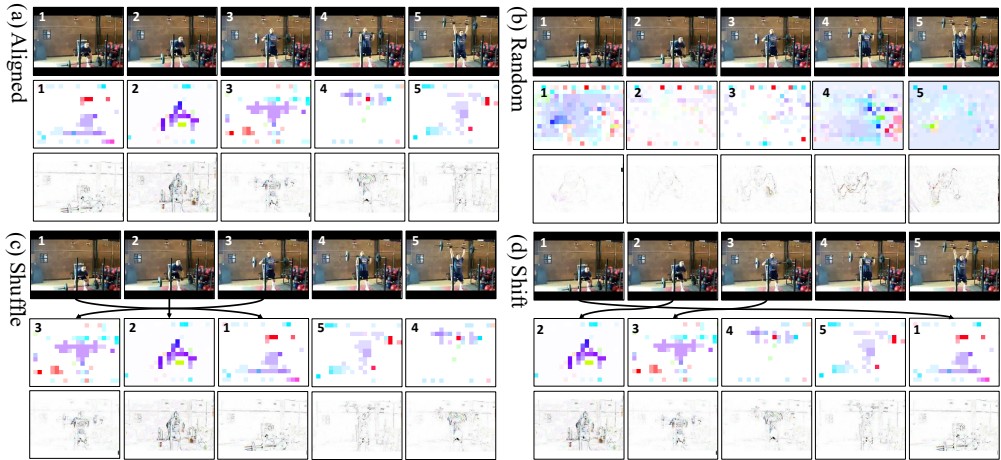

Figure 4: **Correspondence type prediction** asks our network to categorize different types of data transformations applied on P-frames. We illustrate four transformation types used in our experiments.

are purposely recorded to highlight important objects and their movements.[1] Therefore, regions with the highest energy of motion can provide clues to learning the desired video representation. We can easily find those regions in compressed videos: the motion vectors in P-frames readily provide magnitude and angular information of motion, which we can harness to find the most vibrant regions.

Based on this intuition, we design a task that asks our model to predict the zeroth-order motion statistics (i.e., the most vibrant region) in a given video. For this, we must be able to deal with a variety of object sizes because a salient moving object can appear at any location in any size. A classical solution to this is to perform pyramidal prediction (Grauman & Darrell, 2005; Lazebnik et al., 2006): We divide a video into spatio-temporal 3D grids at multiple scales and ask our network to predict the most vibrant region at each scale.

Specifically, we define a pyramidal classification task with the following loss function,

$$\mathcal{L}_{PMSP} = -\sum_i \sum_r \sum_q y_{q,r}^{(i)} \cdot \log \alpha_r \left( \mathbf{x}_{q,r}^{(i)} \right) \tag{4}$$

This is a cross-entropy loss computed at every $q$-th grid in every $r$-th level of a spatio-temporal pyramid; $i$ is the sample index. We define a 9-level spatio-temporal pyramid with 3 spatial and 3 temporal scales, i.e., $r \in \{(s,t)|s \in \{[2 \times 2], [3 \times 3], [4 \times 4]\}, t \in \{1, 3, 5\}\}$. The index $q$ iterates over all possible temporal coordinates in the $r$-th level of the pyramid, e.g., in Figure 3 (a), $q \in [0, \cdots, 4]$ with $r = ([2 \times 2], 5)$. $y_{q,r}^{(i)}$ is a one-hot label marking the location with the highest energy of motion in the $q$-th grid in $r$-th level in the pyramid, e.g., in Figure 3 (a), $y_{q,r}^{(i)}$ is a 4-dimensional one-hot vector. We provide a pseudo-code to obtain the ground-truth labels from motion vectors in Appendix. $\mathbf{x}_{q,r}^{(i)}$ is the $(q,r)$-th feature in a 3D grid; we concatenate output embeddings from all three sub-networks, $\mathbf{x}^{(i)} = [\mathbf{x}_I^{(i)}; \mathbf{x}_M^{(i)}; \mathbf{x}_R^{(i)}]$. Finally, $\alpha_r(\cdot)$ is a 2-layer MLP with a softmax classifier predicting the most vibrant region in the given grid; we define one such classifier for each $r$.

**Correspondence Type Prediction (CTP).** One idea often used in self-supervision is applying certain transformations to data and asking a network to predict the correspondence type given a pair of instances (e.g., true pair or randomly selected pair) (Owens & Efros, 2018; Chen et al., 2020; He et al., 2020; Misra & van der Maaten, 2020). The multimodal nature of compressed videos makes them an ideal data format to apply such self-supervision technique: The three frame types in compressed videos exhibit different characteristics, yet they are strongly correlated with each other. This allows us to consider I-frames as a heavily transformed version of the corresponding P-frames, and vice versa. Learning the correspondence type between I-frames and P-frames can therefore encourage our network to learn discriminative representation of videos.

---

[1]This is, of course, a weak hypothesis. But we show some convincing empirical evidence in Appendix.

We define a correspondence type prediction task with the following loss function,

$$\mathcal{L}_{CTP} = - \sum_i \sum_j y_j^{(i)} \cdot \log \beta \left( \mathbf{x}_I^{(i)}, \mathcal{T}(\mathbf{x}_M^{(i)}, \mathbf{x}_R^{(i)}, j) \right) \tag{5}$$

where $i$ is the sample index and $j$ iterates over a set of transformations. $y_j^{(i)}$ is a one-hot label indicating different correspondence types determined by the type of transformation done, and $\mathcal{T}(\cdot, j)$ is a data transformation function that changes the input using the $j$-th transformation. We define four transformation types (see Figure 4): (1) **Aligned** keeps the original input (no transformation), (2) **Random** replaces the data with P-frames from a randomly selected video, (3) **Shuffle** randomly shuffles the GOP order, (4) **Shift** randomly divides GOPs into two groups and switch the order, e.g., $[1, 2, 3, 4, 5]$ to $[2, 3, 4, 5, 1]$. Finally, $\beta(\cdot)$ is a 2-layer MLP with a softmax classifier.

Note that there is a nuanced difference between random P-frames and shuffled/shifted P-frames. The former contains P-frames that come from a different clip, while the latter contains P-frames of the same clip as the I-frames, yet in a different frame order. Intuitively, the former encourages our network to learn from global (clip-level) correspondence, while the latter formulates a local (frame-level) correspondence task. Therefore, our CTP task encourages our network to learn discriminative representations at both global and local levels. We provide empirical evidence showing the importance of this global-local mixed objective in Section 3.2.

**Final Objective.** We optimize our model using a learning objective $\mathcal{L}_{PMSP} + \lambda \mathcal{L}_{CTP}$ with $\lambda = 1$. The classifiers $\alpha_r$ and $\beta$ are used only during self-supervised training; we detach them thereafter.

## 3 EXPERIMENTS

**Implementation Detail.** We adopt 3D ResNet (He et al., 2016) as the backbone; see Appendix for architectural details. We establish bidirectional dynamic connections after {*conv1, res2, res3, res4*} layers. We pretrain our model end-to-end from scratch for 20 epochs, including the initial warm-up period of 5 epochs. For downstream scenarios, we finetune our model for 500 epochs for UCF-101 and for 300 epochs for HMDB-51, including the warm-up period of 30 epochs. For both the pretraining and finetuning stages, we use SGD with momentum 0.9, weight decay $10^{-4}$, and half-period cosine learning rate schedule. We use 4 NVIDIA Tesla V100 GPUs and use a batch size of 100.

**Data.** We pretrain our model on Kinetics-400 (Kay et al., 2017). For evaluation, we finetune the pretrained model for action recognition using UCF-101 (Soomro et al., 2012) and HMDB-51 (Kuehne et al., 2011). We use 2-second video clips encoded in 30 FPS with a GOP size $T = 12$. We use all $T = 5$ GOPs but subsample every other P-frames within each GOP; this results in 5 I-frames and 25 P-frames. We randomly crop $224 \times 224$ pixels from videos resized to 256 pixels in the shorter side while keeping the aspect ratio. For data augmentation, we resize the video with various scales $[.975, .9, .85]$ and apply random horizontal flip. For test videos, we take three equidistant $224 \times 224$ pixel crops from videos resized to 256 pixels to fully cover the spatial region. We approximate the fully-convolutional testing (Wang et al., 2018) by averaging the softmax scores for final prediction.

### 3.1 SUPERVISED LEARNING EXPERIMENTS

We first demonstrate our proposed IMR network in the fully-supervised setup, training it without using our self-supervised pretext tasks. We use the standard training and evaluation protocols for both UCF-101 (Soomro et al., 2012) and HMDB-51 (Kuehne et al., 2011). For fair comparisons with existing approaches (Wu et al., 2018; Shou et al., 2019), we report results both when we train the model from scratch and when we pretrain it on Kinetics-400 (Kay et al., 2017) and finetune it on downstream datasets (indicated in column `Pretrain`).

Table 1 summarizes the results. When trained from scratch, our model outperforms CoViAR (Wu et al., 2018) by a large margin regardless of the chosen backbone. The performance gap is alleviated when the models are pretrained on Kinetics400, but our approach continues to outperform them even in this scenario. This suggest that CoViAR struggles to learn discriminative representations without help from a large-scale pretraining data. We believe the performance gap comes from the difference in how the two models encode compressed videos: CoViAR combines information from I-frames and P-frames only after encoding them separately, while we combine them in the early layers of CNN.

| Models | OF | Pretrain | Backbone | UCF101 | HMDB51 |
|---|---|---|---|---|---|
| CoViAR[‡] | ✗ | Scratch | ResNet152 | 43.8 | 27.3 |
| IMR (No connection) | ✗ | Scratch | 3D-ResNet18 | 52.7 | 34.6 |
| IMR (Unidirectional) | ✗ | Scratch | 3D-ResNet18 | 69.7 | 40.8 |
| IMR (No conv) | ✗ | Scratch | 3D-ResNet18 | 71.7 | 42.6 |
| IMR (No attention) | ✗ | Scratch | 3D-ResNet18 | 73.2 | 43.5 |
| IMRNet | ✗ | Scratch | 3D-ResNet18 | 74.1 | 43.7 |
| IMRNet | ✗ | Scratch | 3D-ResNet50 | 80.2 | 55.9 |
| CoViAR[†] | ✗ | ImageNet | ResNet152 (I), ResNet18 (P) | 90.4 | 59.1 |
| CoViAR[‡] | ✗ | Kinetics400 | ResNet152 | 90.8 | 59.2 |
| IMRNet (Ours) | ✗ | Kinetics400 | 3D-ResNet18 | 91.4 | 62.8 |
| IMRNet (Ours) | ✗ | Kinetics400 | 3D-ResNet50 | **92.6** | **67.8** |
| CoViAR[†] | ✓ | ImageNet | ResNet152 (I), ResNet18 (P, OF) | 94.9 | 70.2 |
| DMC-Net[†] | ✓ | ImageNet | ResNet152 (I), ResNet18 (P) | 90.9 | 62.8 |
| DMC-Net[†] | ✓ | ImageNet | ResNet152 (I), I3D (P) | 92.3 | 71.8 |
| IMRNet (Ours) | ✓ | Kinetics400 | 3D-ResNet50 (I, P), I3D (OF) | **95.1** | **72.2** |

Table 1: **Results from the supervised setting.** Column `OF` indicates results using optical flow during training. Column `Pretrain` indicates datasets used for supervised pretraining. [†]: published results. [‡]: our results based on official implementations by the authors.

| Models | ResNet152[*] | R(2+1)D[†] | CoViAR[‡] | DMC[‡] | IMR[‡] (R18) | IMR[‡] (R50) |
|---|---|---|---|---|---|---|
| Preprocess (ms) | 75.00 | 75.00 | 2.87 | 2.87 | 2.87 | 2.87 |
| Inference (ms) | 7.50 | 1.74 | 1.30 | 1.91 | 1.36 | 1.44 |
| Total (ms) | 82.50 | 76.74 | 4.17 | 4.78 | 4.23 | 4.31 |
| GFLOPs | 11.3 | 0.96 | 4.2 | 4.4 | 0.66 | 1.04 |

Table 2: **Runtime analysis** of per-frame speed (ms) and FLOPs. The number of input frames are different across models: [*] 1 frame (since it is a 2D CNN), [†] 16 frames, [‡] 25 frames.

CoViAR and DMC-Net reported improved results when they are trained using optical flow. Therefore, we also conduct experiments by adding an I3D network (Carreira & Zisserman, 2017) to encode optical flow images; we simply concatenate our IMRNet features with the I3D features as our final representation (no lateral connections between IMRNet and I3D). This model outperforms both CoViAR and DMC-Net trained with optical flow (bottom group, Table 1). DMC-Net improves upon CoViAR by adapting GANs (Goodfellow et al., 2014) to reconstruct optical flow from P-frames. Note that our approach (with 3D-ResNet50 backbone) outperforms DMC-Net (with ResNet152/18 backbones) on both datasets *even without using optical flow* during training and thus significantly simplifies the training setup (no GANs required).

Next, we conduct an ablation study on the bidirectional dynamic connection: (a) **No connection** removes lateral connections and thus is similar to CoViAR, (b) **Unidirectional** establishes connections from M/R-Networks to I-Network, but not vice versa, i.e., Equation equation 3 becomes $\hat{x}_M = x_M, \hat{x}_R = x_R$, (c) **No conv** replaces (de-)conv layers with simple up/down-sampling, (d) **No attention** removes the multimodal-gated attention module. The results are shown in Table 1. We can see that lateral connections are critical component of our model (`Ours` vs. `No connection`) and doing so in a bidirectional fashion significantly improves performance (`Ours` vs. `Unidirection`). We can also see that using (de-)conv layers and dynamically modulating the connection with gate functions improve performance (`Ours` vs. `No conv` and `No attention`).

Table 2 shows *per-frame* runtime speed (ms) and GFLOPs measured on an NVIDIA Tesla P100 GPU with Intel E5-2698 v4 CPUs (∗ process individual frames. † and ‡ process 16- and 25-frame sequences, respectively). Our approach has the same preprocessing time of CoViAR and DMC because all three approaches use the same video loader implementation (Wu et al., 2018). As for the inference speed, IMRNet is comparable to CoViAR and even slightly faster than DMC (we divide the total inference time by #frames following the convention of Wu et al. (2018)). This is partly because we use lighter backbones (R18/R50 vs. R152 used in CoViAR and DMC) to compensate for the expensive 3D convolutional operations, while DMC requires an OF generator network of 7 all-convolutional layers, which adds extra cost. In terms of per-frame FLOPs, ours is more efficient than CoViAR and DMC because the computation is done at the sequence-level rather than per-frame; we observe a similar trend for R(2+1)D (which uses ResNet18) vs. ResNet152. This shows that our

| Models | Compressed | Modality | Pretext | Pretrain | Backbone | UCF101 | HMDB51 |
|---|---|---|---|---|---|---|---|
| C3D | ✗ | V | MotPred | Kinetics400 | C3D | 61.2 | 33.4 |
| 3D-ResNet18 | ✗ | V | RotNet3D | Kinetics600 | 3D-ResNet18 | 62.9 | 33.7 |
| 3D-ResNet18 | ✗ | V | ST-Puzzle | Kinetics400 | 3D-ResNet18 | 65.8 | 33.7 |
| R(2+1)D-18 | ✗ | V | ClipOrder | UCF101 | R(2+1)D-18 | 72.4 | 30.9 |
| 3D-ResNet34 | ✗ | V | DPC | Kinetics400 | 3D-ResNet34 | 75.7 | 35.7 |
| Multisensory | ✗ | A+V | Multisensory | Kinetics400 | Audio-VisualNet | 82.1 | – |
| AVTS | ✗ | A+V | AVTS | Audioset | MC3 | 89.0 | 61.6 |
| ELo | ✗ | A+V | ELo | Kinetics400 | (2+1)D ResNet-50 | 93.8 | 67.4 |
| CoViAR‡ | ✓ | V | Scratch | None | ResNet152 | 43.8 | 27.3 |
| IMRNet | ✓ | V | Scratch | None | 3D-ResNet18 | 74.1 | 43.7 |
| CoViAR‡ | ✓ | V | AOT | Kinetics400 | ResNet152 | 53.6 | 29.3 |
| CoViAR‡ | ✓ | V | Rotation | Kinetics400 | ResNet152 | 56.7 | 31.4 |
| IMRNet | ✓ | V | InfoNCE | Kinetics400 | 3D-ResNet18 | 73.9 | 43.7 |
| IMRNet | ✓ | V | AOT | Kinetics400 | 3D-ResNet18 | 74.6 | 44.0 |
| IMRNet | ✓ | V | Rotation | Kinetics400 | 3D-ResNet18 | 75.1 | 44.3 |
| CoViAR‡ | ✓ | V | PMSP | Kinetics400 | ResNet152 | 63.5 | 35.9 |
| CoViAR‡ | ✓ | V | CTP | Kinetics400 | ResNet152 | 64.4 | 37.4 |
| CoViAR‡ | ✓ | V | CTP (Binary) | Kinetics400 | ResNet152 | 63.7 | 37.1 |
| IMRNet | ✓ | V | PMSP | Kinetics400 | 3D-ResNet18 | 76.0 | 44.9 |
| IMRNet | ✓ | V | CTP | Kinetics400 | 3D-ResNet18 | 76.7 | 44.8 |
| IMRNet | ✓ | V | CTP (Binary) | Kinetics400 | 3D-ResNet18 | 74.6 | 44.2 |
| IMRNet | ✓ | V | PMSP+CTP | Kinetics400 | 3D-ResNet18 | **76.8** | **45.0** |

Table 3: **Results from the self-supervised setting.** Column `Compressed` indicates the methods that learn directly from compressed videos without decoding them. `Modality` indicates whether a method used only visual (**V**) modality or audio-visual modalities (**A+V**). `Pretrain` indicates datasets used for self-supervised pretraining. ‡: based on an official implementation by the authors.

3D CNN backbones do not bring any significant extra cost compared to CoViAR and DMC, and thus our model enjoys all the computational benefits of compressed video processing.

## 3.2 Self-supervised Learning Experiments

We move to the self-supervised regime and demonstrate our pretext tasks by pretraining our IMRNet on Kinetics400 (Kay et al., 2017) and transferring it to action recognition. Because ours is the first self-supervised approach to learn compressed video representation, there exist no published baseline that we can directly compare with. Therefore, we provide results from existing self-supervised approaches that require the decoding step. We include approaches that learn from RGB images – **AOT** (Wei et al., 2018), **Rotation** (Jing et al., 2018), **MotPred** (Wang et al., 2019a), **RotNet3D** (Jing et al., 2018), **ST-Puzzle** (Kim et al., 2019), **ClipOrder** (Xu et al., 2019), **DPC** (Han et al., 2019) – as well as those that learn from audio and visual channels in videos – **Multisensory** (Owens & Efros, 2018), **AVTS** (Korbar et al., 2018), **Elo** (Piergiovanni et al., 2020).

Table 3 summarizes the results. We first notice that pretraining the models with any pretext tasks improves downstream performance (the first group of results), suggesting self-supervised pretraining is effective in general. We also see that IMRNet pretrained using our pretext tasks (PMSP+CTP) outperforms the baseline pretext tasks (second group) and self-supervised methods for uncompressed videos (third group). This shows the effectiveness of our IMRNet pretrained with our pretext tasks.

Next, we conduct an ablation study by pretraining the base models using either PMSP and CTP alone. We also test `CTP (Binary)` which is a simplied version of our CTP task with only two modes: Aligned and Random (see Figure 4). Note that this is a typical pair correspondence setup used in the literature (Arandjelovic & Zisserman, 2017). Table 3 (fourth group) shows the results. We can see that using either of our pretext tasks leads to a significant improvements compared to the `Scratch` result. The `CTP (Binary)` results suggests that the two additional transformation types (Shuffle and Shift in Figure 4) improves the task by making it more difficult to solve; we noticed that the loss curve of `CTP (Binary)` decreases significantly faster than `CTP` and quickly saturates thereafter.

## 4 RELATED WORK

**Self-supervised learning of video representation.** Self-supervised learning has received significant attention (Kumar BG et al., 2016; Santa Cruz et al., 2017; Doersch et al., 2015; Wang & Gupta, 2015). Based on strong progress in the image domain, several works proposed to learn video representations in a self-supervised manner. One popular idea is leveraging temporal information (Wang & Gupta, 2015; Isola et al., 2015; Jayaraman & Grauman, 2016; Misra et al., 2016; Fernando et al., 2017; Wei et al., 2018). Temporal coherence of video pixels has been leveraged as a self-supervisory signal (Vondrick et al., 2018; Wang et al., 2019c). Another popular idea is learning transformation-invariant representations (Kim et al., 2019; Gidaris et al., 2018; Jing et al., 2018). Also, contrastive learning (Oord et al., 2018; Hjelm et al., 2018; He et al., 2020; Chen et al., 2020) has been successfully applied to videos (Han et al., 2019). Despite active research in this field, to the best of our knowledge, there has not been prior work on self-supervised learning from compressed videos.

**Compressed video recognition.** Compressed video understanding has been tackled in a supervised setting (Zhang et al., 2016; Wu et al., 2018; Shou et al., 2019). Existing approaches encode each stream separately and perform late fusion, *e.g.*, feature concatenation (Zhang et al., 2016; Wu et al., 2018). However, as we show in our experiments, this can miss out useful information that can only be learned by modeling the interaction across streams. Unlike previous approaches, our approach shares relevant information across streams during the encoding process. In addition, because compressed videos do not provide continuous RGB frames, it is not easy to directly apply 3D CNNs to encode I-frames. Therefore, existing approaches use 2D CNNs to process compressed video frames, e.g., CoViAR (Wu et al., 2018) uses 2D CNNs to process each stream and perform average pooling over P-frames, which is insufficient to model complex motion dynamics. DMC-Net (Shou et al., 2019) reconstructs the optical flow from P-frames and later use the reconstructed signal as input to I3D (Carreira & Zisserman, 2017), but this requires ground-truth optical flow which is compute-intensive. Instead, our IMR network adopts the gated attention Hu et al. (2018); Ryoo et al. (2020a) and bidirectional connection Ryoo et al. (2020b); Feichtenhofer et al. (2019) for lateral connection to model complex motion dynamics with I,P frames freely available in compressed videos.

## 5 CONCLUSION

We introduced an IMR network for compressed video recognition and two pretext tasks for self-supervised learning of compressed video representation. Our work complements and extends existing work on compressed video recognition by (1) proposing the first self-supervised training approach on the compressed videos, and (2) proposing a three-stream 3D CNN architecture to encode compressed videos while dynamically modeling interaction between I-frames and P-frames. We demonstrated that our IMRNet outperforms state-of-the-art approaches for compressed videos in both fully-supervised and self-supervised settings, and that our pretext tasks yield better performance in downstream tasks.

**Acknowledgement** We thank SNUVL lab members, especially Hyeokjun Kwon and Hyungyu Park, for their helpful discussions. This research was supported by Seoul National University, Brain Research Program by National Research Foundation of Korea (NRF) (2017M3C7A1047860), and AIR Lab (AI Research Lab) in Hyundai Motor Company through HMC-SNU AI Consortium Fund, and Institute of Information & communications Technology Planning & Evaluation (IITP) grant funded by the Korea government (MSIT) (No.2019-0-01082, SW StarLab).

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

# A  PMSP GROUND-TRUTH LABELS

We obtain the ground-truth labels for the pyramidal motion statistics prediction (PMSP) task directly from motion vectors provided in compressed videos. Algorithm 1 shows a pseudo-code to compute the labels at multiple spatio-temporal scales, $r \in \{(s, t) | s \in \{[2 \times 2], [3 \times 3], [4 \times 4]\}, t \in \{1, 3, 5\}\}$.

---

**Algorithm 1:** Self-supervision label for Pyramidal Motion Statistics Prediction

---

[1] Motion vectors $\{M_{0,1}, ..., M_{T,K}\}$ with $T$ GOPs each having $K - 1$ motion vectors Generate
  $dx, dy$ by convolving motion vectors with Prewitt operator, $G_x, G_y$ $t \leftarrow 1$ ;
  ▷ Set temporal scale t to 1 $Y \leftarrow []$ ▷ Empty list for labels $i = 0$ 2 $n = 0$ $t - 1$
$sum\_dx \leftarrow \text{sum}(dx[n * K : (n + T - t + 1) * K])$
$sum\_dy \leftarrow \text{sum}(dy[n * K : (n + T - t + 1) * K])$
$magnitude \leftarrow \text{cartToPolar}(sum\_dx, sum\_dy)$
$magnitude_{[2 \times 2]} \leftarrow \text{makeGrid}(magnitude, spatial = 2)$
$magnitude_{[3 \times 3]} \leftarrow \text{makeGrid}(magnitude, spatial = 3)$
$magnitude_{[4 \times 4]} \leftarrow \text{makeGrid}(magnitude, spatial = 4)$
$y_{[2 \times 2]} \leftarrow \arg\max_{q \in [1, \cdots, 2^2]}(magnitude_{[2 \times 2]})$ $y_{[3 \times 3]} \leftarrow \arg\max_{q \in [1, \cdots, 3^2]}(magnitude_{[3 \times 3]})$
$y_{[4 \times 4]} \leftarrow \arg\max_{q \in [1, \cdots, 4^2]}(magnitude_{[4 \times 4]})$ $Y \leftarrow Y \cup [y_{[2 \times 2]}, y_{[3 \times 3]}, y_{[4 \times 4]}]$ $t \leftarrow t + 2$
PMSP labels, $Y$, at multiple scales $\{(s, t) | s \in \{[2 \times 2], [3 \times 3], [4 \times 4]\}, t \in \{1, 3, 5\}\}$

---

# B  PMSP LABEL VISUALIZATION

In Section 2.3 of the main paper, we motivated the design of our PMSP task by arguing that there is an implicit *videographer bias* captured in videos in-the-wild that naturally reflects visual saliency: Videos are purposely recorded to highlight important objects and their movements; therefore, regions with the highest energy of motion – captured by our PMSP labels – can provide clues to learning video representation that captures salient moving objects. We acknowledged that this is, of course, a weak hypothesis (footnote 1 in the main paper). However, in this section we provide some convincing empirical evidence.

Figures 5-12 are generated by visualizing regions with the highest energy of motion – *i.e.* , the PMSP labels – at multiple spatio-temporal scales. It needs a bit of explanation on how to read the figures as there is a lot going on. Each figure is organized into three rows; each row shows results with multiple spatial regions at a particular temporal scale, $t \in \{1, 3, 5\}$. We color-code different spatial scales: Red boxes are in a $[2 \times 2]$ spatial scale, green boxes are in a $[3 \times 3]$ spatial scale, and blue boxes are in a $[4 \times 4]$ spatial scale. Notice that all five I-frames in the top rows ($t = 1$) in each set of results always contain identical regions. This is because, at the temporal scale $t = 1$ (meaning, a temporal grid of size 1), we compute the regions with the highest motion energy over the entire video (5 GOPs), hence the regions are identical across all I-frames in a video. Conversely, the bottom rows ($t = 5$, a temporal grid of size 5) show the regions computed at each GOP, and hence the regions may differ by every I-frame (recall that each GOP contains a single I-frame). The middle rows ($t = 3$) show regions computed over 3 GOPs. We overlay the regions at the overlapping I-frames, *e.g.*, the third I-frame at $t = 3$ contains regions computed at all three grid locations spanning over the I-frame indices [1,2,3], [2,3,4], and [3,4,5]. We order the figures at an increasing level of complexity, and provide detailed analyses of the results in the captions of the figures.

The results in Figures 5-12 suggest that the most vibrant regions, as computed by our PMSP labels, tend to overlap with *semantically* important regions, *e.g.*, the most salient moving objects. Intuitively, training our model to detect those regions encourages it to learn visual representations that capture salient objects and motion. This allows our model to learn discriminative visual representation in a self-supervised manner.

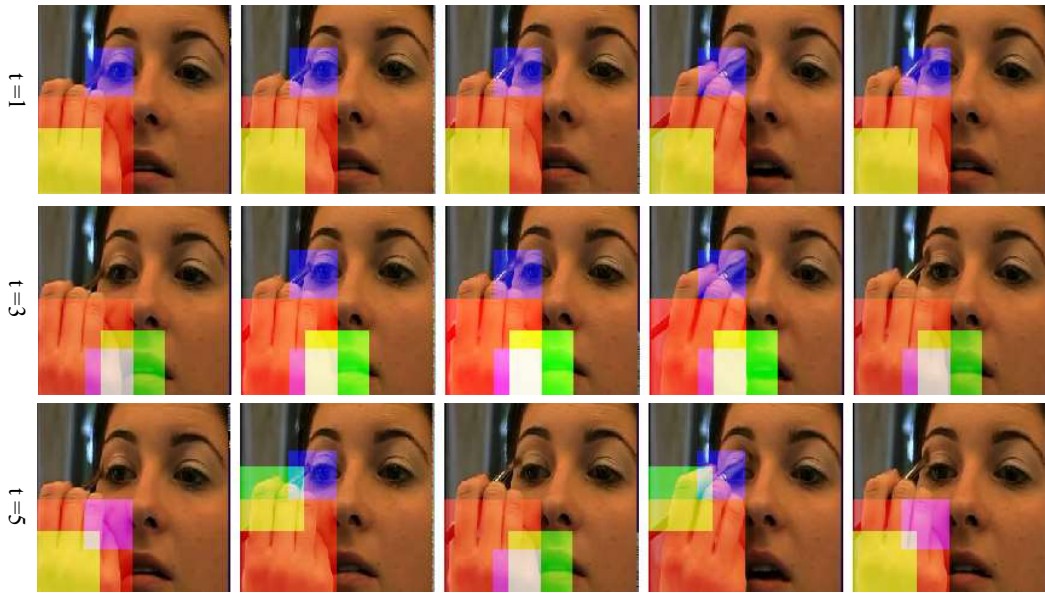

Figure 5: **PMSP label visualization.** The most vibrant regions, as highlighted by boxes of varying sizes indicating different spatial scales ([2 × 2], [3 × 3], [4 × 4]), all successfully capture the most salient moving object (the hand with a eye brush) and its motion (applying eye makeup).

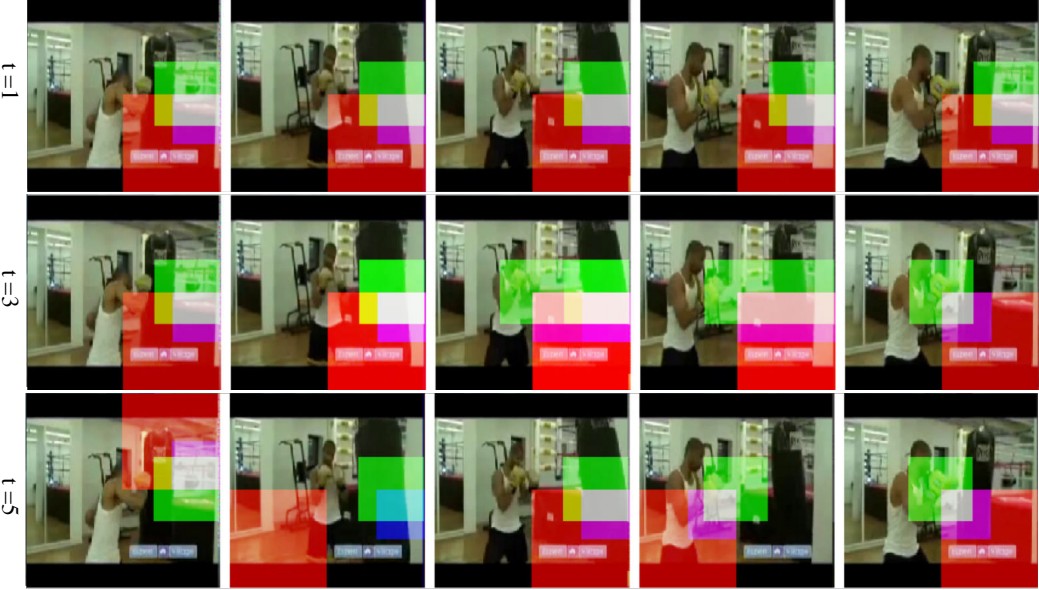

Figure 6: **PMSP label visualization.** At $t = 1$, the most vibrant region is the punching bag, correctly capturing the semantic category of the video (*BoxingPunchingBag*). As we get temporally finer, the regions start to capture the boxer's movement, *e.g.*, the elongated green boxes in the third and the fourth I-frames at $t = 3$.

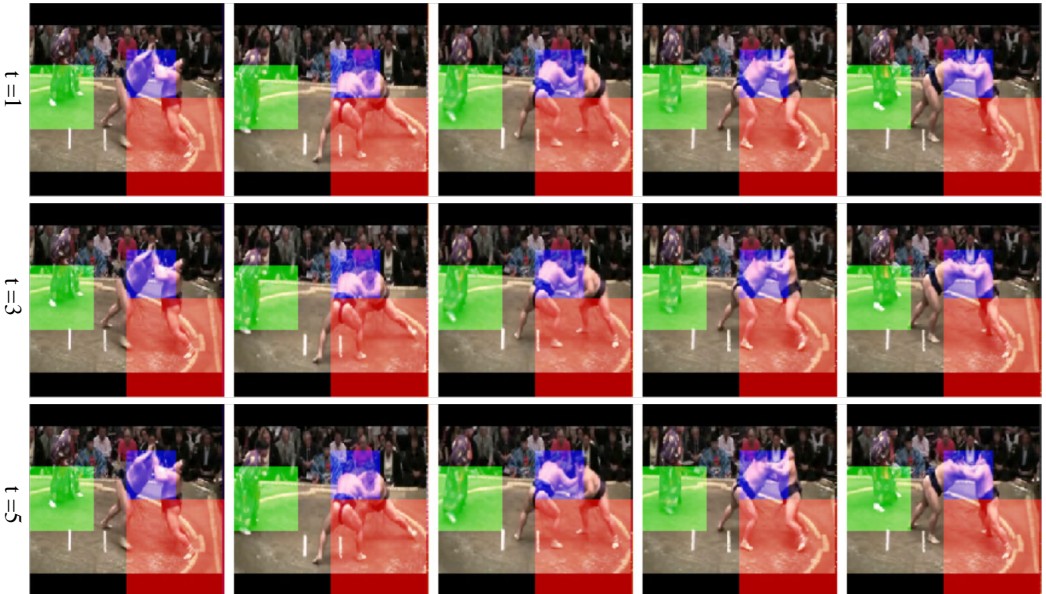

Figure 7: **PMSP label visualization.** This example highlights the benefit of formulating our task in a *spatially* pyramidal manner. Notice the boxes at different spatial scales capture different moving objects at varying sizes, i.e., green boxes ($[3 \times 3]$) capture the referee, blue boxes ($[4 \times 4]$) capture the hands of the two sumo wrestlers, and red boxes ($[2 \times 2]$) capture the wrestlers' legs.

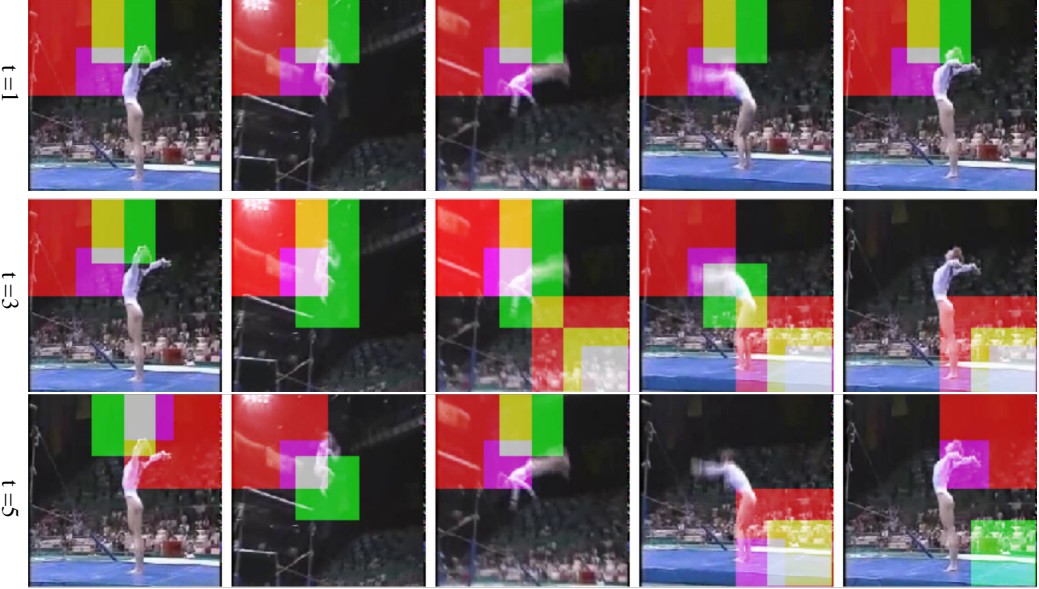

Figure 8: **PMSP label visualization.** This example highlights the benefit of formulating our task in a *temporally* pyramidal manner. While the vibrant regions at $t = 1$ fail to capture the tumbling gymnast, the vibrant regions at $t = 3$ and $t = 5$ successfully track her trajectory (especially the three middle I-frames at $t = 3$). In general, different videos will contain moving objects at different speeds; our pyramidal formulation allows us to capture a wide variety of moving objects at different speeds via multiple temporal scales.

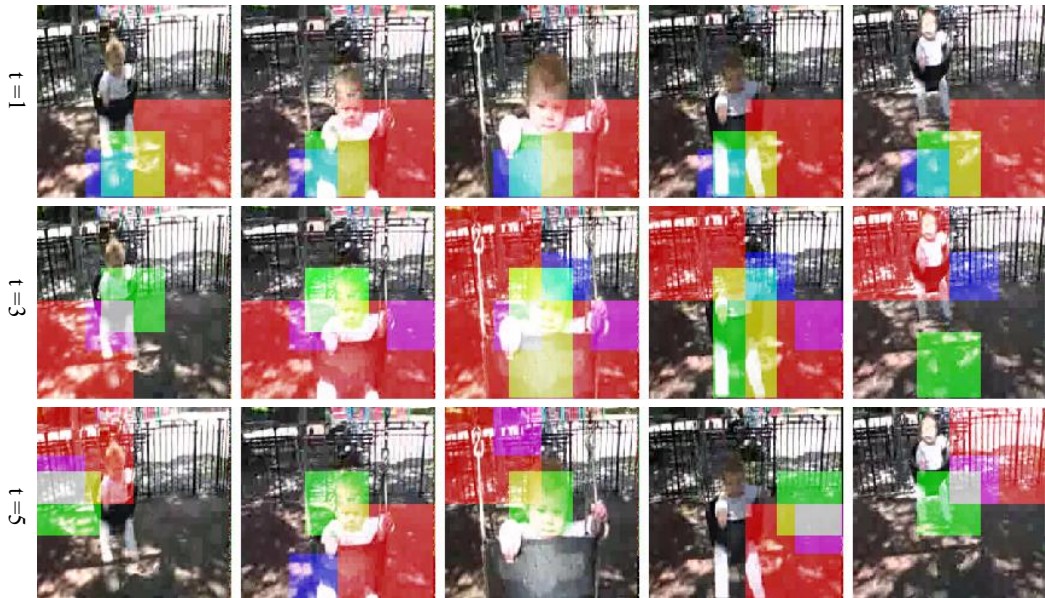

Figure 9: **PMSP label visualization.** Another example highlighting the benefit of our pyramidal formulation. While some regions at $t = 1$ miss the toddler in swing (see the first and the fifth I-frames in the first row), at $t = 3$ and $t = 5$ the boxes successfully track the toddler's trajectory.

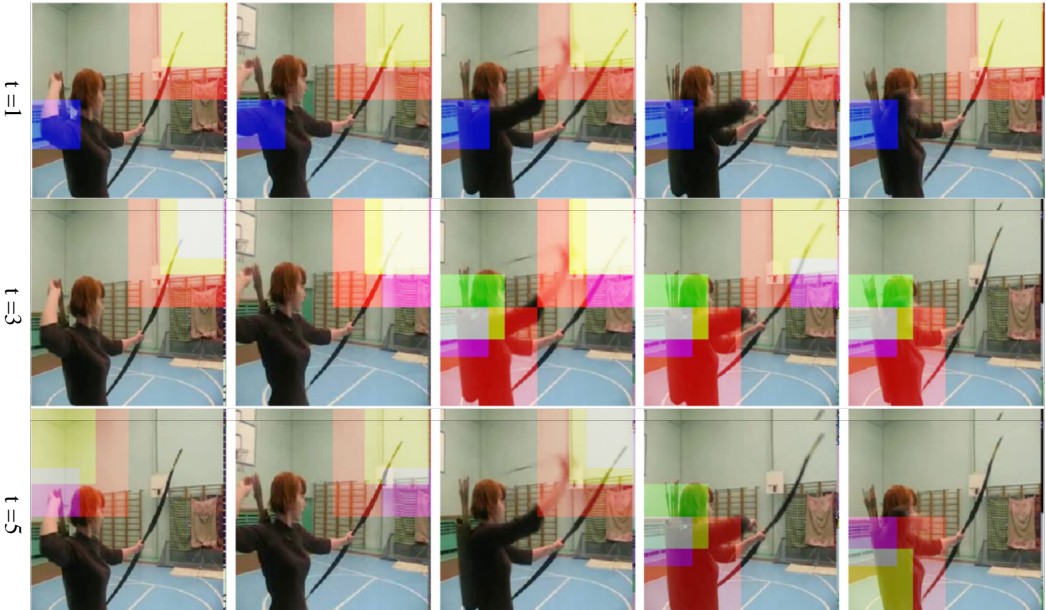

Figure 10: **PMSP label visualization.** This example contains a dynamically moving object (a women with a bow and an arrow) spanning across a large region in the frames, representing a challenging situation. Notice the boxes at different spatial and temporal scales highlight different parts: at $t = 1$, both the green ($[3 \times 3]$) and the red ($[2 \times 2]$) boxes capture the bow, which exhibits the sharpest edge with motion (hence the highest motion energy in those spatial scales), while the blue boxes ($[4 \times 4]$) capture the arm that takes out an arrow. At $t = 3$ and $t = 5$, the regions start to capture different parts, *e.g.*, the woman, which exhibits dynamic motion only towards the end of the video.

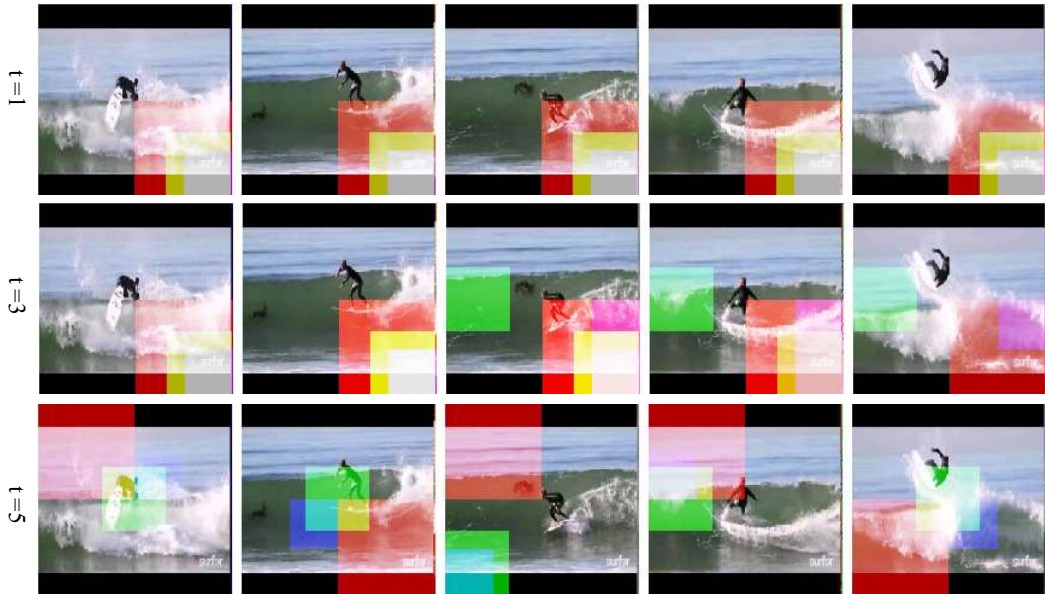

Figure 11: **PMSP label visualization.** Another challenging example containing a small, dynamically moving object (Surfing). At $t = 1$, all the boxes focus on the crushing waves on the bottom right corner, which is on average the most vibrant region in this video. Things are not much better at $t = 3$; the surfer is still too fast to capture, and thus the boxes fail to capture the surfer and instead highlight crushing waves. At the finest temporal scale $t = 5$, the boxes begin to capture the surfer (see the first, second and the fifth frames on the bottom).

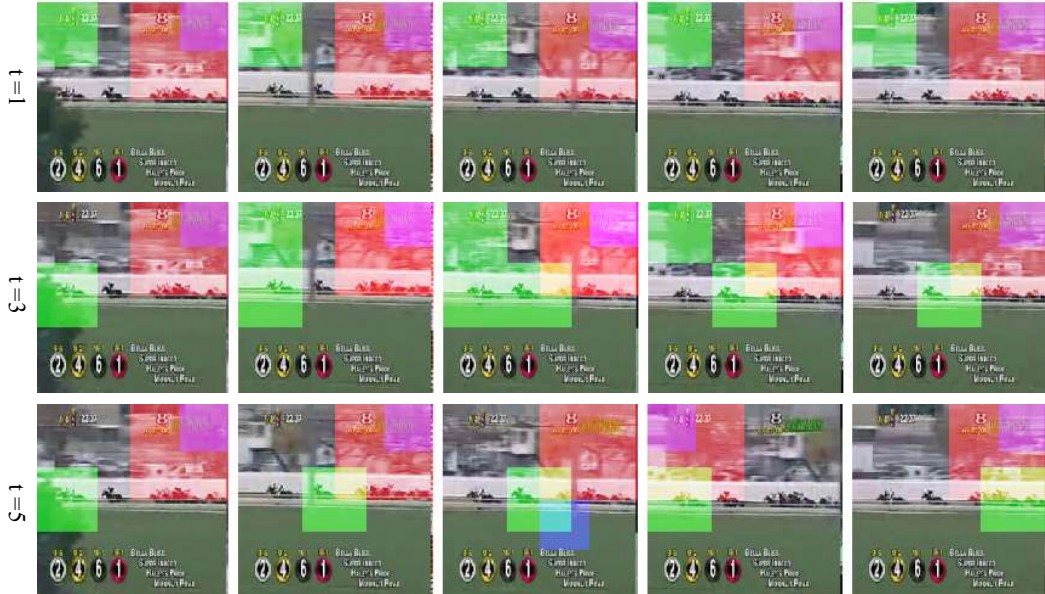

Figure 12: **PMSP label visualization.** This is a *partial*-failure example that shows boxes highlighting game statistics during horse race TV broadcast (see the boxes at $t = 1$). The game statistics constantly change frame-by-frame (*e.g.*, time marks, rank, etc.), which caused those regions to exhibit on average the highest energy of motion for the entire duration of the video. Learning representations that strictly focus on those regions could lead to non-discriminative information (many sports videos show similar game statistics on screen). Fortunately, the boxes begin to highlight the horse riders at a finer temporal scale; see the green boxes ($[3 \times 3]$) at $t = 3$ and $t = 5$. This, again, suggests that the pyramidal formulation makes our PMSP task robust to a variety of challenging real-world scenarios.

## C  PMSP PREDICTION RESULTS

Figure 13 and Figure 14 show side-by-side comparisons of the ground-truth PMSP labels and our prediction results. We adopt the same visualization scheme used in the figures in Appendix B. Overall, our prediction results are mostly identical to the ground-truth regions. When our prediction deviates from the ground-truth, the predicted regions still tend to capture important moving objects, e.g., Figure 13 (a) captures different parts of the punching bag, Figure 14 (b) captures different parts of the boxer, and Figure 14 (d) captures different arms of the swimmer.

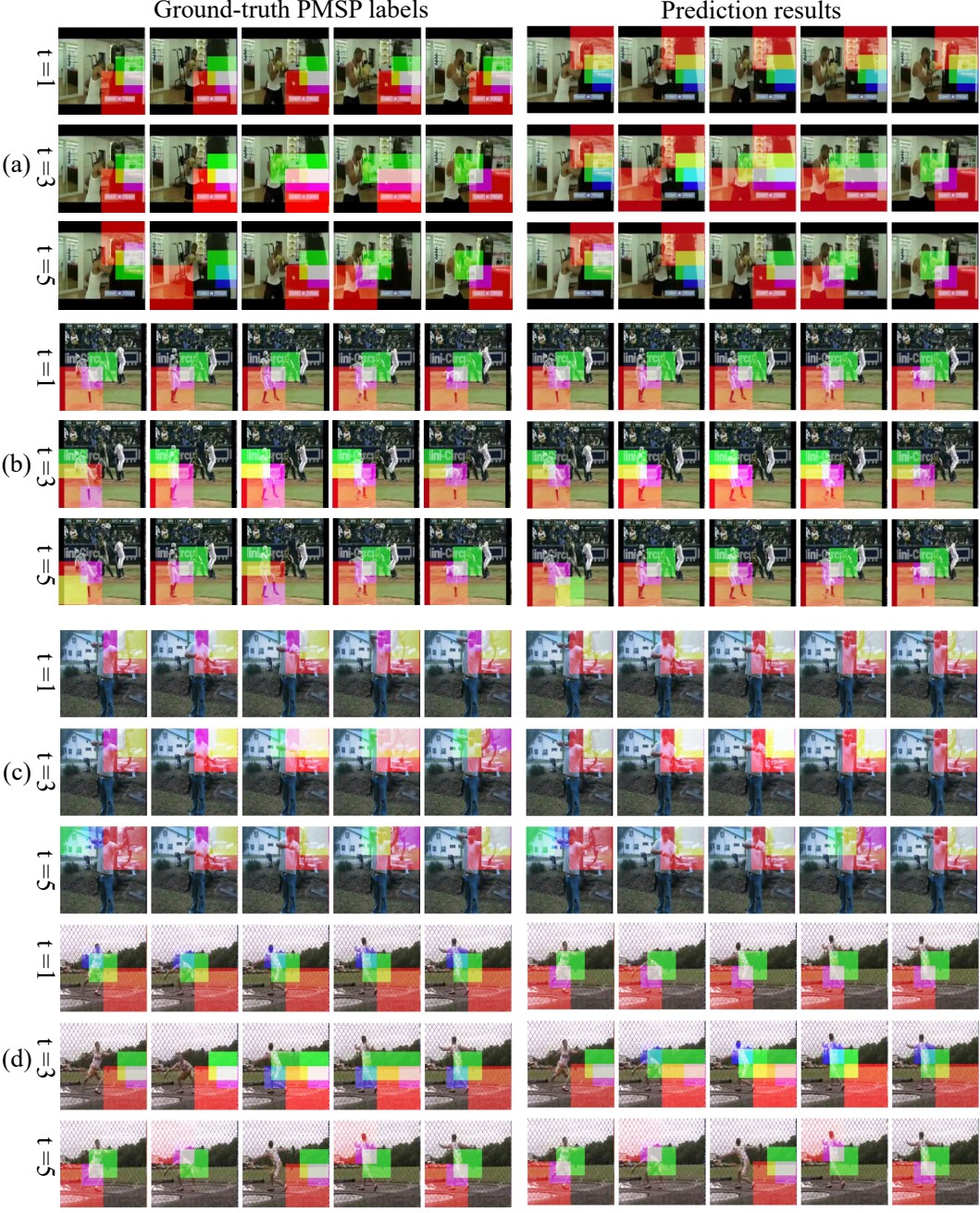

Figure 13: **PMSP prediction results.** Overall, the predicted regions tend to highlight salient moving objects (although sometimes different from the ground-truth). (a): BoxingPunchingBag, (b): BaseballPitch, (c): Archery, (d): ThrowDiscus.

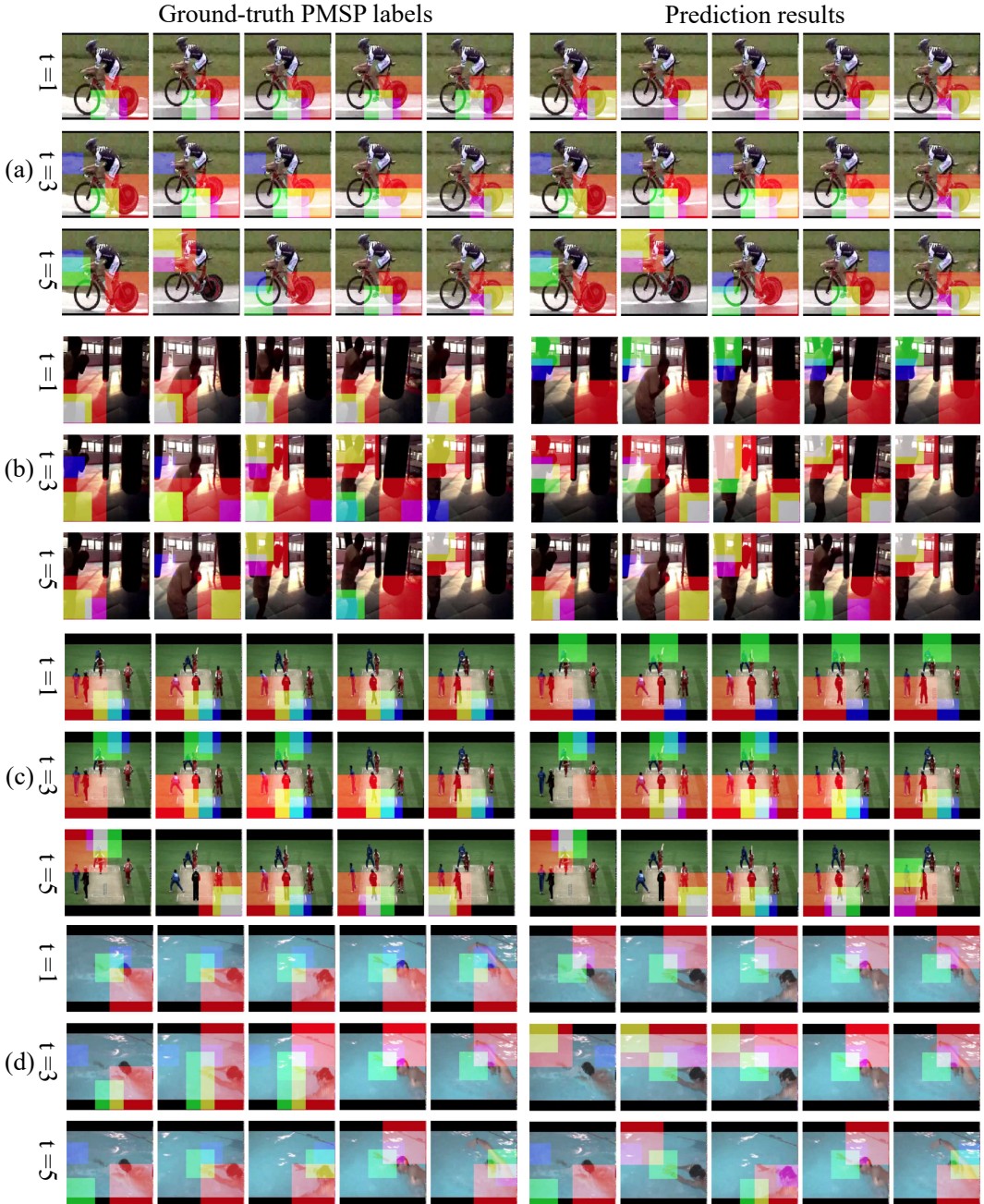

Figure 14: **PMSP prediction results.** Overall, the predicted regions contain the salient moving objects (although sometimes different from the ground-truth). (a): Biking, (b): BoxingPunchingBag, (c): CricketShot, (d): FrontCrawl.

# D   VIDEO-TO-VIDEO RETRIEVAL

To demonstrate the quality of video representations learned using our self-supervised learning objectives (Section 2.3 in the main paper), we evaluate our method in the video-to-video retrieval task. To do this, we measure the cosine similarity between a query video and all the other video in a candidate set, and show the top-1 retrieved video. We compare ours to two baselines: `3D Rotation` Jing et al. (2018) is our IMRNet pretrained using the 3D rotation prediction task (we used the *IMRNet + Rotation* pretrained model reported in Table 2 of our main paper), and `ImageNet` is a ResNet152 fully-supervised with ImageNet ILSVRC-2012 Russakovsky et al. (2015). We visualize the results in Figures 15-18 and analyze the results in the caption of each figure.

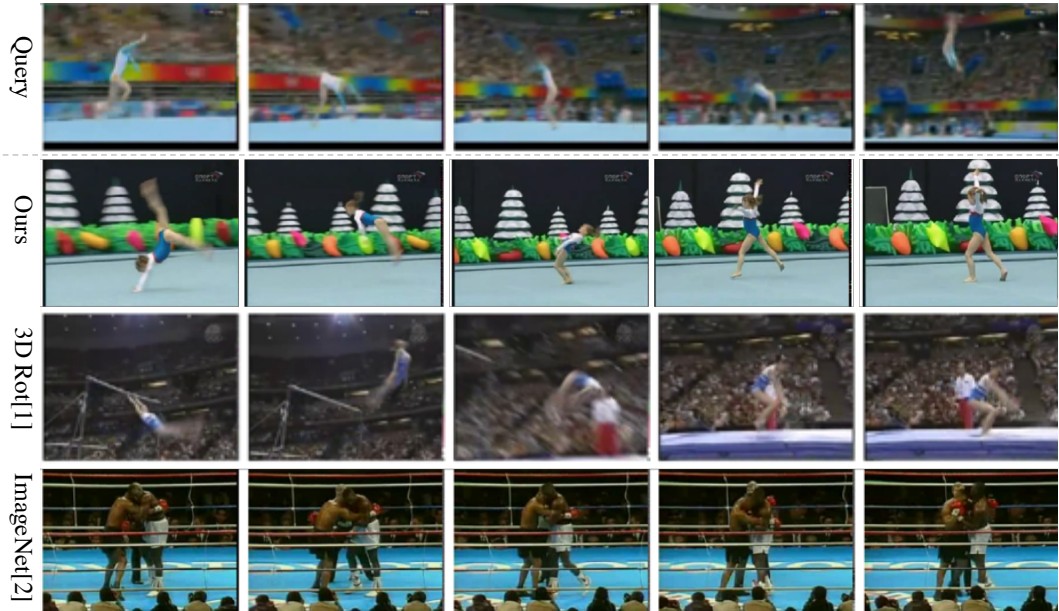

Figure 15: **Video-to-video retrieval result.** Ours finds the most similar video to the query in terms of both the appearance (a gymnast) and the motion (handspring). The 3D Rotation baseline captures perhaps more similar appearance (a gymnast with the audience in the back) but less similar motion (horizontal bar jump vs. handspring). The ImageNet baseline fails to capture both appearance and motion (ImageNet does not contain a category relevant to floor gymnastics).

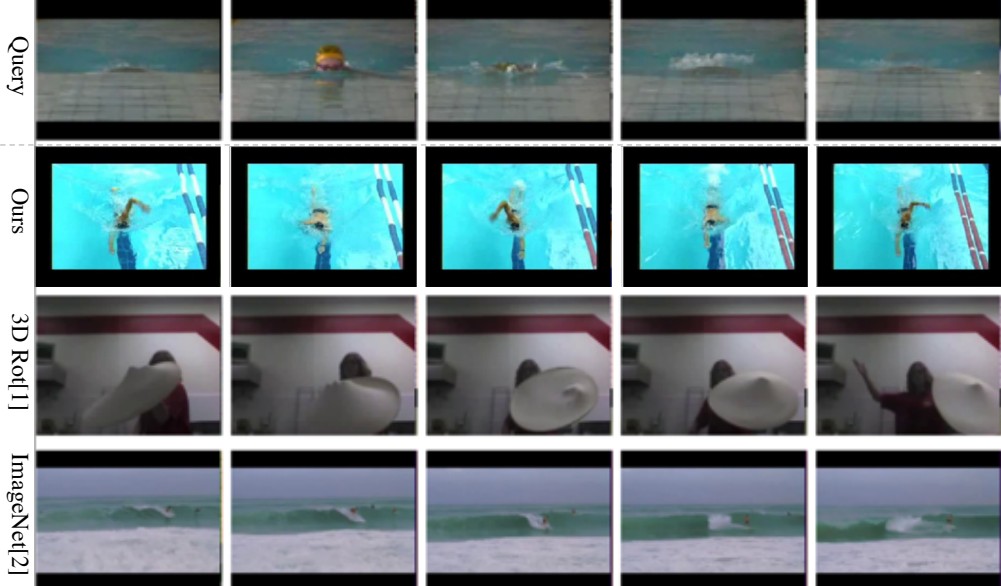

Figure 16: **Video-to-video retrieval result.** Ours finds the most similar video to the query in terms of both the appearance (swim stadium) and motion (swimming). The ImageNet baseline does capture similar appearance (water), but fails to capture motion (swimming vs. surfing). The 3D Rotation baseline shows little to no semantic similarity to the query video.

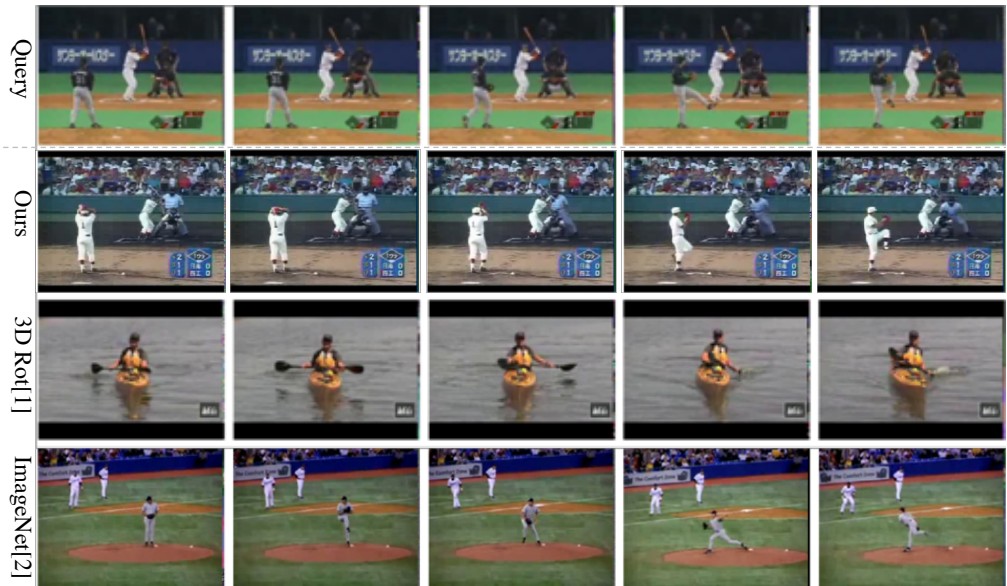

Figure 17: **Video-to-video retrieval result.** Ours finds the most similar video to the query in terms of both the appearance (scene layout) and the motion (pitching). The ImageNet baseline does capture similar high-level semantics appearance-wise (baseball pitcher) but motion is relatively less similar (different camera angle, no catcher and no hitter). The 3D Rotation baseline shows little to no semantic similarity to the query video.

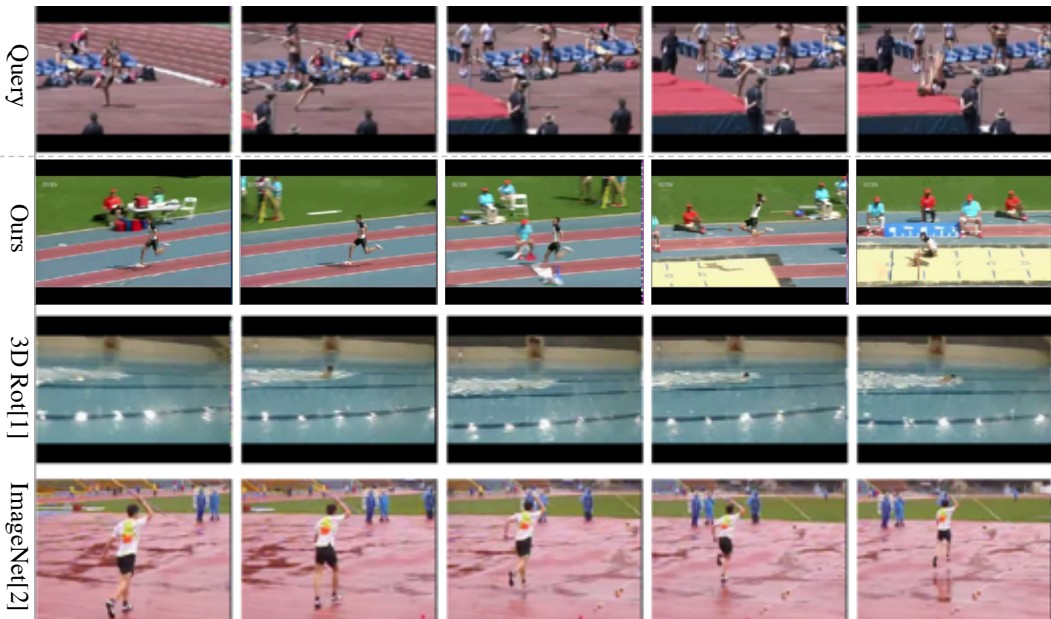

Figure 18: **Video-to-video retrieval result.** All three retrieval results fail to find videos that belong to the same semantic category as the query video (pole vault). However, ours finds a video that contains similar appearance (running track) and similar motion (running and jumping). The ImageNet baseline also captures similar appearance (javelin throw) but less similar motion (running at a substantially slower pace). The 3D Rotation baseline shows little to no semantic similarity to the query video.

# E  ARCHITECTURE DETAILS

Table 4 provides architecture details of our IMRNet. In our experiments, we used both 3D ResNet-18 and 3D ResNet-50 as the backbone; we provide the details of both models in the table. We also include the details of our bidirectional dynamic connections, which include 3D convolutional/deconvolutional layers that downsamples/upsamples the computed features along the temporal dimension. We establish the connections after {*conv1, res2, res3, res4*} layers, each with different numbers of channels.

| Stage | I Pathway | M/R Pathway | Output sizes $T \times S^2$ |
|---|---|---|---|
| raw clip | – | – | $60 \times 224^2$ |
| data layer | stride $12, 1^2$ | stride $2, 1^2$ | I: $5 \times 224^2$ 
 M/R: $25 \times 224^2$ |
| $conv_1$ | $1 \times 7^2, 64$ 
 stride $1, 2^2$ | $5 \times 7^2, 8$ 
 stride $1, 2^2$ | I: $5 \times 112^2$ 
 M/R: $25 \times 112^2$ |
| $pool_1$ | $1 \times 3^2, max$ 
 stride $1, 2^2$ | $5 \times 3^2, max$ 
 stride $1, 2^2$ | I: $5 \times 56^2$ 
 M/R: $25 \times 56^2$ |
| $res_2$ | (3D ResNet-18) 
 $\begin{bmatrix} 1 \times 3^2, 64 \\ 1 \times 3^2, 64 \end{bmatrix} \times 2$ 
 (3D ResNet-50) 
 $\begin{bmatrix} 1 \times 1^2, 64 \\ 1 \times 3^2, 64 \\ 1 \times 1^2, 256 \end{bmatrix} \times 3$ | (3D ResNet-18) 
 $\begin{bmatrix} 3 \times 3^2, 4 \\ 1 \times 3^2, 4 \end{bmatrix} \times 2$ 
 (3D ResNet-50) 
 $\begin{bmatrix} 3 \times 1^2, 4 \\ 1 \times 3^2, 4 \\ 1 \times 1^2, 16 \end{bmatrix} \times 3$ | I: $5 \times 56^2$ 
 M/R: $25 \times 56^2$ |
| $res_3$ | (3D ResNet-18) 
 $\begin{bmatrix} 1 \times 3^2, 128 \\ 1 \times 3^2, 128 \end{bmatrix} \times 2$ 
 (3D ResNet-50) 
 $\begin{bmatrix} 1 \times 1^2, 128 \\ 1 \times 3^2, 128 \\ 1 \times 1^2, 512 \end{bmatrix} \times 4$ | (3D ResNet-18) 
 $\begin{bmatrix} 3 \times 3^2, 8 \\ 1 \times 3^2, 8 \end{bmatrix} \times 2$ 
 (3D ResNet-50) 
 $\begin{bmatrix} 3 \times 1^2, 8 \\ 1 \times 3^2, 8 \\ 1 \times 1^2, 32 \end{bmatrix} \times 4$ | I: $5 \times 28^2$ 
 M/R: $25 \times 28^2$ |
| $res_4$ | (3D ResNet-18) 
 $\begin{bmatrix} 3 \times 3^2, 256 \\ 1 \times 3^2, 256 \end{bmatrix} \times 2$ 
 (3D ResNet-50) 
 $\begin{bmatrix} 3 \times 1^2, 256 \\ 1 \times 3^2, 256 \\ 1 \times 1^2, 1024 \end{bmatrix} \times 6$ | (3D ResNet-18) 
 $\begin{bmatrix} 3 \times 3^2, 16 \\ 1 \times 3^2, 16 \end{bmatrix} \times 2$ 
 (3D ResNet-50) 
 $\begin{bmatrix} 3 \times 1^2, 16 \\ 1 \times 3^2, 16 \\ 1 \times 1^2, 64 \end{bmatrix} \times 6$ | I: $5 \times 14^2$ 
 M/R: $25 \times 14^2$ |
| $res_5$ | (3D ResNet-18) 
 $\begin{bmatrix} 3 \times 3^2, 512 \\ 1 \times 3^2, 512 \end{bmatrix} \times 2$ 
 (3D ResNet-50) 
 $\begin{bmatrix} 3 \times 1^2, 512 \\ 1 \times 3^2, 512 \\ 1 \times 1^2, 2048 \end{bmatrix} \times 3$ | (3D ResNet-18) 
 $\begin{bmatrix} 3 \times 3^2, 32 \\ 1 \times 3^2, 32 \end{bmatrix} \times 2$ 
 (3D ResNet-50) 
 $\begin{bmatrix} 3 \times 1^2, 32 \\ 1 \times 3^2, 32 \\ 1 \times 1^2, 128 \end{bmatrix} \times 3$ | I: $5 \times 7^2$ 
 M/R: $25 \times 7^2$ |

| Stage | $conv_1$ | $res_2$ | $res_3$ | $res_4$ |
|---|---|---|---|---|
| I to M/R | $1 \times 5^2, 8$ 
 stride $5, 1^2$ | $1 \times 5^2, 8$ 
 stride $5, 1^2$ | $1 \times 5^2, 16$ 
 stride $5, 1^2$ | $1 \times 5^2, 32$ 
 stride $5, 1^2$ |
| M/R to I | $5 \times 7^2, 4$ 
 stride $5, 1^2$ | $5 \times 7^2, 4$ 
 stride $5, 1^2$ | $5 \times 7^2, 8$ 
 stride $5, 1^2$ | $5 \times 7^2, 16$ 
 stride $5, 1^2$ |

Table 4: **IMRNet architecture details**. We show two versions of IMRNet with different backbones: 3D ResNet-18 and 3D ResNet-50. We denote the input dimensions by *{temporal size, spatial size²}*, kernels by *{temporal size, spatial size², channel size}* and strides by *{temporal stride, spatial stride²}*.

