# OpenReview forum: "Self-Supervised Learning of Compressed Video Representations"
_ICLR.cc/2021/Conference — ICLR 2021 Poster_

### Official Review · AnonReviewer2 · 2020-10-15

**Rating:** 6
**Confidence:** 5

**Review:**

This paper proposes an approach to self-supervised learning from videos. The approach takes advantage of compressed videos, using the encoded residuals and motion vectors within the video codec. Using encoded videos has been shown to reduce computation time required by decoding videos. Previous works have explored compressed videos for supervised recognition, showing the potential, while this paper introduces a way to leverage compressed videos for self-supervised learning.

The paper is well written and easy to follow. The proposed IMRNet shows benefit over previous supervised compressed video approaches. The paper proposes two self-supervised tasks related to compressed videos: one to predict the spatial locations with movement and one to predict the temporal matching of the inputs (Correspondence Type Prediction).

The Correspondence Type Prediction tasks are fairly standard, and have been explored by many previous works (which are cited). The main difference being the use of the compressed video features instead of rgb frame or audio.

An interesting observation is that the two tasks when combined don't seem to be any better, roughly 0.1%-0.3%, which is likely within noise on smaller datasets like HMDB and UCF101.

The comparisons in Table 3 are missing comparisons to state-of-the-art approaches. For example,
- "Cooperative learning of audio and video models from self-supervised synchronization", NeurIPS'18
- "Audio-visual scene analysis with self-supervised multisensory features", ECCV'18
- "Evolving Losses for Unsupervised Video Representation Learning", CVPR'20

all outperform this approach. While those approaches use audio features and are not specifically focused on compressed videos, there's nothing about them making them incompatible with compressed videos using say only I-frames. I think Table 3 could be improved by comparing to existing approaches for non-compressed videos as well as using those approaches on compressed videos. This would help show the benefit of the proposed tasks.

The paper has some missing details. One of the claims is that compressed videos is faster, which is shown in Table 2. But there are some missing details, for example, the number of training epochs is not reported (only that 30 are used for warmup). How many does this use? How much faster is this method than non-compressed method?

There are also some other related works, e.g.
"AssembleNet: Searching for Multi-Stream Neural Connectivity in Video Architectures", ICLR'20
proposes a similar idea to the bidirectional connections and
"AssembleNet++: Assembling Modality Representations via Attention Connections", ECCV'20
proposes ideas similar to the multimodal gated attention.


Overall, the paper is interesting. I think some of the experiments could be strengthen to better compare to existing self-supervised approaches.

---

> ### Author Response · Authors · 2020-11-22
> **Thank you! We incorporated your thoughtful suggestions to our revision**
>
> We appreciate your positive comments and thoughtful feedback.
>
> * **Missing references in Table 3:** We have added the three SoTA results in Table 3, indicating that they all use an additional audio modality for learning and are based on decoded RGB frames. While we feel that these baselines wouldn’t necessarily be the most direct comparison to ours, the results show that there is still room for improvements.
>
> * **Missing details**
>
>     * We pretrained our model for 20 epochs including the warm-up period of 5 epochs. We then finetuned our model for 500 epochs on UCF-101 and for 300 epochs on HMDB-51, including a warm-up period of 30 epochs.
>
>     * Table 2 reports FLOPs and per-frame speed of both the preprocessing step and the inference step, following the convention in the compressed video understanding literature, e.g., CoViAR and DMC. As we can see, the biggest speed difference comes from the preprocessing step (which measures how long it takes to load the video data onto the memory): the two non-compressed models, i.e., ResNet152 and R(2+1)D, take 26x longer than the compressed-based models. This is because the non-compressed models need to decode RGB frames on-the-fly.
>
> * **AssembleNet and AssembleNet++:** We have added both references to our revised paper.

---

### Official Review · AnonReviewer4 · 2020-10-29
**a fair paper yet not surprisedly interesting**

**Rating:** 6
**Confidence:** 5

**Review:**

This paper is not bad in the sense that:
1. important problem -- how to leverage compressed video such a natural and efficient format is very important while under-explored by the community
2. extensive experiments -- clear gains on standard benchmarks demonstrate the proposed method's gains. good ablation studies in like table 3 demonstrate the effectiveness of each proposed pretext task.
3. the paper explored the how to better fuse multiple streams and two pretext ssl tasks i.e. PMSP+CTP, which all turn out to be successful

Yet, the way fusing multiple streams using techniques like multi-modal attention and the pretext tasks i.e PMSP and CTP -- they look fair while do not surprise me. It is quite natural to think of these techniques to improve the current models. For example, correspondence has been used by UCB researchers to do SSL in video yet for other task.

But I do acknowledge it takes significant work to validate these findings and conduct such comprehensive explorations and comparisons. I do believe these findings are valuable to the broad video community and thus will have good impact. Thus, I lean to accept this paper.

---

> ### Author Response · Authors · 2020-11-22
> **Appreciate the positive (even though somewhat lukewarm) feedback!**
>
> Thank you for the thoughtful comments and a recommendation to accept our paper! We understand somewhat lukewarm sentiment towards our algorithmic treatments to exploit the inherent structure of compressed videos. As the first attempt to self-supervised compressed video understanding, we wanted to use simple and intuitive techniques to clearly show the benefit of our approach. We appreciate the comment that the findings of our work will be “valuable to the broad video community and thus will have a good impact.” We do hope that our results will encourage the community to follow up with effective and more novel algorithms to bring further improvements.

---

### Official Review · AnonReviewer1 · 2020-11-01
**Good topic but the approach lacks novelty**

**Rating:** 6
**Confidence:** 4

**Review:**

Summary:
The authors study a new problem – self-supervised learning for compressed video understanding. It can dramatically save the compute and storage requirements for action recognition. The experiment results on several datasets demonstrate that the proposed approach can achieve the best results over the baselines.

Strength:
1. Self-supervised compressed video understanding is an exciting topic. It can save computation and storage for online video understanding.
2. The two pretext tasks are efficient and straightforward for compressed video understanding.

Questions:

1. Figure 2 is misleading, T_I is k times smaller than T_M,3D Deconv should be on the top and 3D Conv should be on the bottom; the image cubes are also incorrect -- they are supposed to be placed in H-W-C plane. Please re-organize the text and the figures to make the main text clear.
2. The authors design two pretext tasks according to the heuristic: pyramidal motion statistics prediction and correspondence type prediction. I want to say the authors have done a fair evaluation, but the approach's novelty is limited. I would like to see more discussion about the intuition of the proposed method.
3. Why not compare with the contrastive learning-based approaches for self-supervised learning. We can still get the positive pairs from the same video, though it is difficult to augment the  videos give the compressed videos.

---

> ### Author Response · Authors · 2020-11-22
> **Thanks! We clarified novelty & provided more discussion about the intuition**
>
> We appreciate your constructive comments and an acknowledgement that our work studies “a new problem” and “an exciting topic” of self-supervised compressed video understanding.
>
> * **Figure 2:**
>
>     * Thanks for pointing out that the Deconv and 3D Conv blocks are misplaced. This was our mistake and we have fixed it in the revision.
>
>     * The image cubes are actually correct; we had a coordinate sign specifying that the cubes are in the T-HW-C plane. The updated figure now highlights the coordinate system in red.
>
> * **Novelty & intuition:** Our novelty mainly comes from the way we exploit the inherent structure of compressed videos for self-supervised representation learning. As our response to R4 below indicated, we wanted to use simple and intuitive techniques to clearly show the benefit of our ideas. The reviewer is making a great point that providing a discussion about intuition is important for this. Below we provide more insights about the PMSP and CTP tasks.
>
>     * The PMSP task exploits the videographer bias -- i.e., videos are purposely recorded to highlight important objects and their movements. The PMSP task encourages our network to learn discriminative representations that help capture the most vibrant regions in the spatio-temporal space. We note that Sec. B in the Appendix provides extended discussion about the intuition of PMSP with empirical evidence; we kindly ask the reviewer to revisit Sec.B and C in the Appendix, especially the captions of Figures 5-12, as they provide comprehensive insights about the task.
>
>     * The CTP task largely follows the intuition of correspondence prediction often used in self-supervised learning, including contrastive learning approaches (we also mentioned this in the paper). However, our formulation is slightly nuanced in that the negative P-frames not only come from different videos, i.e., Fig. 4(b) Random -- as is typically done in the literature, e.g., [Korbar et al., 2018, Owens & Efros 2018] among others -- but also come from the same clip in a different frame order, i.e., Fig. 4(c) Shuffle and 4(d) Shift. The intuition is that providing negative P-frames of the same clip will encourage the network to learn more discriminative representations based on local (frame-level) correspondence, while the first type only provides global (clip-level) correspondence signals. We believe that the use of “Shuffle” and “Shift” negatives has not been explored before (though its technical novelty is not overwhelmingly high). We have added this discussion in the paper to emphasize the nuanced difference.
>
> * **Contrastive baseline:** Thank you for this great suggestion! We evaluated the contrastive baseline using the vanilla InfoNCE loss used in SimCLR [Chen et al., 2020], taking negatives by randomly shuffling the embeddings of the I-frames and the P-frames within the same batch. This achieved 73.9% on UCF-101 and 43.7% in HMDB51 (we have added the new results in Table 3).  We note that this baseline is somewhat similar to the CTP (Binary) baseline (in Table 3) in that both methods use negative P-frames from a different clip; the only meaningful difference is that one is multi-class classification and another is a binary classification. Our new result confirms the importance of adding the “Shuffle” and “Shift” negatives (Fig 4).

---

> > ### Comment · AnonReviewer1 · 2020-11-22
> > **'image cube' not the 'feature cube'**
> >
> > Thanks for the response. I have to say the authors convince me with the clarification and the comparison experiments with contrastive learning based approach. The figures look much better to me. Please revisit the figures and make sure there are no
> >  such problems in them any more since those mistakes can mislead the readers and leave a bad impression to them.
> >
> > Although the novelty of the proposed approaches still seems like limited to me. The authors indeed evaluate a practical problem and I would like to raise my score in the discussion period. However, I will reserve my concern about the novelty and I will not be uncomfortable if the chair decide not to accept the paper.
> >
> > Regarding to the 'image cube'. I did notice the T-HW-C plane. I mean that the T dimension of the image cube (input) seems like the C dimension of the following feature cube. Obviously, T is the depth of the feature cube, but T is the width of the image cube. This is a minor issue and will not make so much difference to help readers understand the pipeline.

---

> > > ### Author Response · Authors · 2020-11-22
> > > **We revised Fig. 2 and caption again (ever so slightly)**
> > >
> > > Thanks for further clarification! Now we see where the confusion comes from -- we changed where the T dimension appears between the input cubes and the feature cubes. We did this to visually emphasize how C changes at different stages of inference. We have revised Fig 2 and caption to clarify that the T-HW-C coordinate system is for the "feature tensors (blue and orange cubes)" to avoid further confusion.

---

### Decision · Program_Chairs · 2021-01-07
**Final Decision**

**Decision:**

Accept (Poster)

**Comment:**

Reviewers liked the self-supervised learning of compressed videos, noting that it is an "exciting topic" and an "important problem", although they found the proposed methods (PMSP andCTP) less exciting. Reviewers were satisfied with the execution and the extensive experimental studies. AC felt the community may benefit from the paper's intuitive integration of self-supervised learning and the compressed video's signals (I and P frames, residuals, motion vectors, etc).